# Multi-Factorized Semi-Covariance of Stock Markets and Gold Price

**Yun Shi [1], Lin Yang [2,*], Mei Huang [3] and Jun Steed Huang [4]**

[1] School of Management, Bay Campus, Swansea University, Swansea SA1 8EN, UK; sherry.shi@visionx.org
[2] Department of Mathematics, Shanghai University of Finance and Economics, Shanghai 200433, China
[3] Department of Management, University of Toronto, Toronto, ON, M1C 1A4, Canada; mei.huang12505@gmail.com
[4] Department of Computer, Carleton University, Ottawa, ON, K1S 5B6, Canada; steedhuang@ujs.edu.cn
[*] Correspondence: yanglin@163.sufe.edu.cn

**Abstract:** Complex models have received significant interest in recent years and are being increasingly used to explain the stochastic phenomenon with upward and downward fluctuation such as the stock market. Different from existing semi-variance methods in traditional integer dimension construction for two variables, this paper proposes a simplified multi-factorized fractional dimension derivation with the exact Excel tool algorithm involving the fractional center moment extension to covariance, which is a complex parameter average that is a multi-factorized extension to Pearson covariance. By examining the peaks and troughs of gold price averages, the proposed algorithm provides more insight into revealing underlying stock market trends to see who is the financial market leader during good economic times. The calculation results demonstrate that the complex covariance is able to distinguish subtle differences among stock market performances and gold prices for the same field that the two variable covariance may overlook. We take London, Tokyo, Shanghai, Toronto, and Nasdaq as the representative examples.

**Keywords:** fractional moment; stock exchange; complex covariance; semi-variance

## 1. Introduction

Complex algorithms are used in analyzing real-world implementations, as the trusted analytic solution, it typically tends to have challenges in software implementation, is costly, and requires time. Hence, this leads to demand for simpler software solutions by implementing complex theory. Complex theory has received significant interest in recent years and is being increasingly used to solve real-world problems. Among them are combinations of two or more algorithms involving numerical algorithms, analytic calculation, and other computational techniques, such as artificial intelligence, fuzzy systems or simulation, etc. (Zunino et al. 2008; Brooks 1990; Buchanan 2005; Hernandez-Orallo and Dowe 2010; Gerla 2005; Davidovitch Lior and Shtub 2008).

Over the past few decades, scholars have been interested in the relationship between stock prices and the price of gold. For example, during the Global Financial crisis in 2007, the price of gold increased while the stock market crashed and attracted many investors to invest in gold as a security asset (Beckman et al. 2014).

It has been a common belief that gold is a reliable investment and a hedge from risk in the stock markets, especially during crisis. Precious metals such as gold are desirable choices for investors to use as a diversification profile due to their different volatilities from stock prices (Daskalaki and Skiadopoulos 2011). For instance, there were significant cross effects between gold prices and stock prices on return and volatility (Mei and McNown 2019), while gold is a strong safe asset in most developed markets and can effectively reduce the profile risk, especially during the period of financial crisis in China (Arouri et al.

2015). Meanwhile, compared with copper, gold shows a weaker connection to leverage effects, but more sustainable (Hammoudeh and Yuan 2008).

However, even though many studies concur that gold is a hedge and a security in the stock market, gold as a safe haven is not true for all countries. Some scholars believed that there was inter-relationship between the price of gold and stock prices in Vietnam and Thailand, but not in Malaysia, Indonesia, and Philippines (Do and Sriboonchitta 2010). In Baur and McDermott's research in 2010, they indicated that gold is a safe haven in most European countries and the USA, while in Australia, Canada, Japan, and many emerging countries, this is not true. On the other hand, the mixed Clayton–Gumbel model indicates that in the Malaysian market, gold could act as a safe haven. This contradicted Do and Sriboonchitta's 2010 results, where gold could also act as a safe haven in the UK, the USA, Thailand, and Singapore, while in Indonesia, the Philippines, and Japan, the price of gold crashed with the stock market and cannot be used as a security asset (Nguyen et al. 2016). Hence, there are contradictions in the literature and it is clear that the relationship between the price of gold and stock prices vary among different countries.

Apart from geographical effects, time also affects the inter-relationship between precious metal and stock returns while gold was only a hedge for a certain period (Baur and McDermott 2010). Moreover, in the US, the UK, and Germany, gold is a hedge and secure, particularly when the stock market experienced an extreme negative shock effect. Conversely, apart from specific stock events, gold on average is not a safe haven (Baur and Lucey 2010).

Since the early 2000s, the price of gold and stock prices have a reverse relationship, where investors would prefer to buy undervalued stocks when the gold price increases. However, with either extreme low or high volatility, the price of gold did not show a negative relationship with stock prices in the US market (Hood and Malik 2013). A study examines the co-movement across the price of gold and stock prices in Brazil, Russia, India, China, and South Africa. These markets indicate that there is no evidence for any interrelationship between their stock markets and price of gold, which means that gold could act as a safe haven against crisis in these markets (Mensi et al. 2018). However, gold cannot simply be seen as a security asset in investments considering it is a relatively liquid asset and is easy-to-trade. Gold might be suitable for buy low and sell high investors, while for a firm-sized profile, it could bring more negative returns due to the decrease of gold price (Caliskan and Najand 2016).

Many studies have shown that there is some internal connection between the price of gold and stock returns, as even in different markets they can affect each other. For short-term stock returns, it can be predicted by previous gold returns to a certain extent in the US stock market (Baek 2019). In Mei and McNown's study in 2019, they found that the historical returns of the US stock market can be used to predict current returns of gold prices and China's stock market. A full-range tail dependence copula model (Liu et al. 2016) was used to study the relationship between gold and stock returns, and it indicated that there is a significant co-jump and risk-sharing between them (Boako et al. 2019).

Therefore, it can be seen that the relationship between the price of gold and stock returns is a trending topic in the financial world and much research have been done in this area to find the inner connection between them over the past decades. It is also clear that academic research in this area do not agree so far. The main reason is that the relationship is not simply linear, rather it is a nonlinear relation affected by many factors. For example, Africa has a huge gold mine that keeps producing gold with cheaper automated technology. Thailand dealers trade drugs with gold. USA print cash to raise the price of gold. Hence, in this paper, a new method, semi-covariance, will be used to deepen the investigation of the inner nonlinear relationship between the price of gold and stock markets in China, Japan, the UK, Canada, and the USA from the beginning of 1999 to the end of 2019.

Fractal Market Hypothesis (FMH) was first introduced by Edgar E. Peters (1994). Being different from the traditional Efficient Market Hypothesis (EMH), FMH is based on a nonlinear dynamic system, which makes it more reasonable according to the reality. It em-

phasizes the influences on the behaviors of investors caused by the difference of information reception and lengths of investing time horizons or states the existence of fractal structure in stable markets (Xu and Bai 2017). The multiple term covariance (to avoid lengthy matric calculation) is calculated as well to examine the healthy status of the global market for the top four leading countries (Gillard 2019). Cointegration and Copulas (Yolanda et al. 2013) are traditional covariance methods, with strong assumption for Gaussian joint distribution, in reality which may not hold. Lotka–Volterra model (Goran et al. 2019) is good to model non-Gaussian interactions, but the nonlinear calculation is complicated.

However, the FMH and available literatures mentioned methods do not distinguish the rise and fall of the market value. Realized Partial (Co)Variances (Bollerslev et al. 2020) does consider the rise and fall of the market value; but it needs a complicated machine learning algorithm to select the threshold for the rise and fall demarcation point. This paper contributes by focusing on the two separated processes and calculating the natural mean divided semi-covariance that leads to the rise and fall covariance with the price of gold. This article takes the stock markets of China, the United Kingdom, the United States, Japan, and Toronto as examples. After that, this article horizontally compares the correlation between and among the different stock markets and the gold market in another extended dimension. The fundamental objective of this article is to explore the multilateral relationship among the dollar, gold, and stock prices in different countries. In addition, the gold price is based on the U.S. dollar, and the stock index is the day closing value. The analysis can be carried out rapidly by running the Excel tool and the relation of traditional Pearson covariance with the price of gold will be calculated and compared.

## 2. Sub, Super, or Pearson Covariances

The Complex parameter is used as a measure of fluctuation-term memory time series. It relates to the autocorrelations of the time series, and the rate at which these decrease as the semi-covariance lead level between pairs of values increases. Studies involving the Complex parameter were originally developed by Jerome Cardan (1501–1576) (Cardano 1663), used in solving algebra equations.

In order to calculate the Fractional center moment, we need to first generalize the Binomial formula from integer domain to real domain (Zou et al. 2015), where $\xi$ is a random variable whose corresponding expected value is $E\xi$:

$$
\begin{aligned}
\mu_k &= E(\xi - E\xi)^k = E\left( \sum_{i=0}^{\infty} \binom{k}{i} (-1)^i \xi^{k-i} (E\xi)^i \right) \\
&= \sum_{i=0}^{\infty} \binom{k}{i} (-1)^i E(\xi^{k-i})(E\xi)^i. \qquad (1 < k < 3)
\end{aligned}
\tag{1}
$$

When $k = 2$, it is Gauss variance, $k < 2$ is Hypo variance, $k > 2$ is Hyper variance, Factorial of fractional $k$ is calculated by Gamma function below.

We denote that

$$
\binom{k}{i} = \frac{r(r-1)\cdots(r-k-1)}{k!} = \frac{(r)_k}{k!},
\tag{2}
$$

where if $r \in Z$, and $Z$ is the set of all positive integers,

$$
(r)_k = r(r-1)\cdots(r-k-1),
\tag{3}
$$

and if $r \in R$, and $R$ is the set of all real numbers,

$$
(r)_k = \frac{\Gamma(r+1)}{\Gamma(r-k+1)}.
\tag{4}
$$

Note that the Gamma function is defined by:

$$\Gamma(x) = \int_0^\infty t^{x-1} e^{-t} dt. \tag{5}$$

From which we can also have covariance counterpart for two variables or more than two variables (Donald et al. 2012) with the same way for construction called multiple factor correlation (MFC):

$$
\begin{aligned}
\rho_{nk} &= E\left( (z_1 - Ez_1)^k (z_2 - Ez_2)^k \cdots (z_n - Ez_n)^k \right) \\
&= E\left( \prod_{j=1}^n \sum_{i=0}^\infty \binom{k}{i} (-1)^i z_j^{k-i} (Ez_j)^i \right).
\end{aligned}
\tag{6}
$$

When $nk = 2$, it is Pearson covariance, $k < 2$ is Sub covariance, $k > 2$ is Super covariance. In sum, the product that is above the mean will be the real part called upside covariance, and the product that is below the mean will be the imaginary part called downside covariance.

Due to each countries' own holidays, we line up the date first before we calculate the covariance. We noticed that the price of gold is the longest time series when we take the gold date as the baseline. The first rule is: If the day is not a holiday for any of the countries considered here, we run the linear interpolation to generate the price of gold price. The second rule is: If it is a holiday date and it is not on the extended gold list, we delete the day. The third rule is: If the day is a holiday for the country but it is on the extended gold list, we run the linear interpolation to generate the stock value. The fourth rule is: If the holiday lasts more than six days, we delete the middle days.

In order to do a deeper comparison between countries, we further introduce the concept of average positive correlation cycle and negative correlation cycle. We first define the cross point as the day when the upside or downside covariance becomes zero or emerging out of the zeromode period. We count the total numbers of the cross over days and divide the total days considered by the cross over times to obtain the average cycle of the active period. This way we can observe which country has a normal gold to stock relationship and which countries do not.

## 3. Excel Calculations of Semi-Covariances among the Price of Gold and Stock

To use and prove the simplified complex variances, we compare the correlation with the price of gold. Due to Excel's inability to handle the imaginary number, we simplified the calculation with integer power, but separated the upside and downside covariance. We calculated from the beginning of 1999 to the end of 2019, by plotting the curve backwards starting from the most recent years and using the moving window of 10 working pay days, to calculate the covariance averaged over the same period of time to visually smooth out the curve for easier comparisons (Xu et al. 2013).

Figures 1–5 show results for the most fluctuations for the price of gold over the 20 years stock index for China, USA, Japan, UK, and Canada. Figures 6 and 7 are the top four stocks semi-covariance selected based on Table 1, where the first and second half of 10 years covariance average values are shown, the high ones are selected assuming the high ones are dominant global players.

According to statistical principles, the absolute size of covariance can show the strength of the independence of two random variables. In this section, we expand the range of volatility by calculating the semi-covariance between the Shanghai stock market and the price of gold, and keep the two processes of rising and falling, respectively. The first figure shows the relationship between the Shanghai stock index and the price of gold. As seen from the chart, China's stock market has a strong correlation with gold prices for most of 1999–2019 In the index of ordinates, the fluctuation range of the upside covariance is greater than the downside covariance. This means the intensity of a positive relationship is greater than the negative relationship between the price of gold and the Shanghai stock index. In addition, we can observe the calculation result of the semi-covariance of these

two random variables, where above mean is more than below mean in the long period. The probability of positive correlation between the Shanghai stock market and the price of gold is slightly greater than the probability of negative correlation. These results suggest that gold cannot be used as a reasonable hedge in China's financial transactions, as the data show that it fluctuates in the same way as the fluctuation of the stock market. The positive peak day on this curve is 14 February 2008. The reason is that the situation of Shanghai stock market started to be worse at the end of the previous year, and it experienced a sharp rise and fall on that day. These same conclusions were supported by Mensi, Hkiri, Yahyaee, and Kang in 2018.

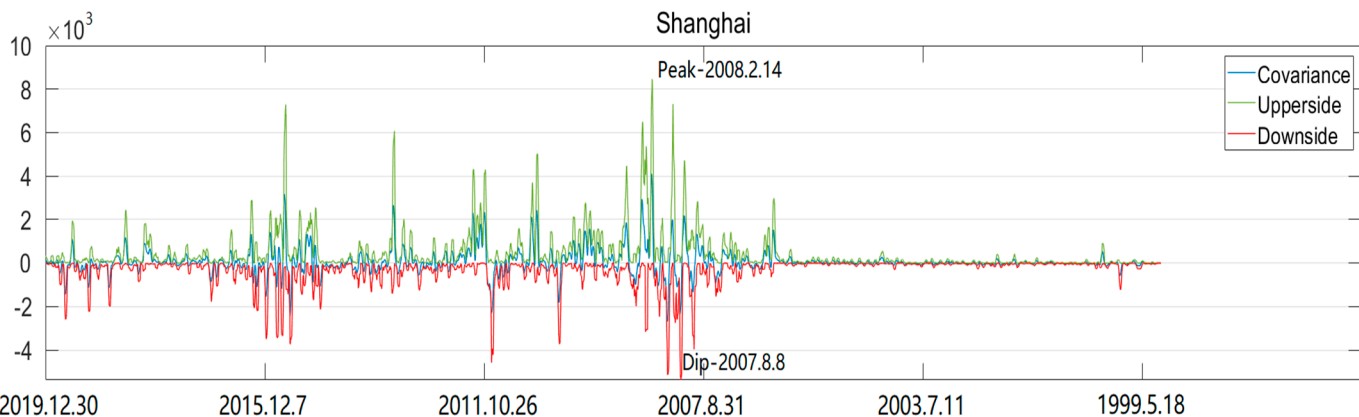

**Figure 1.** Bi-weekly/semi-covariances between gold price and Shanghai Index China from the year 2019 back to 1999.

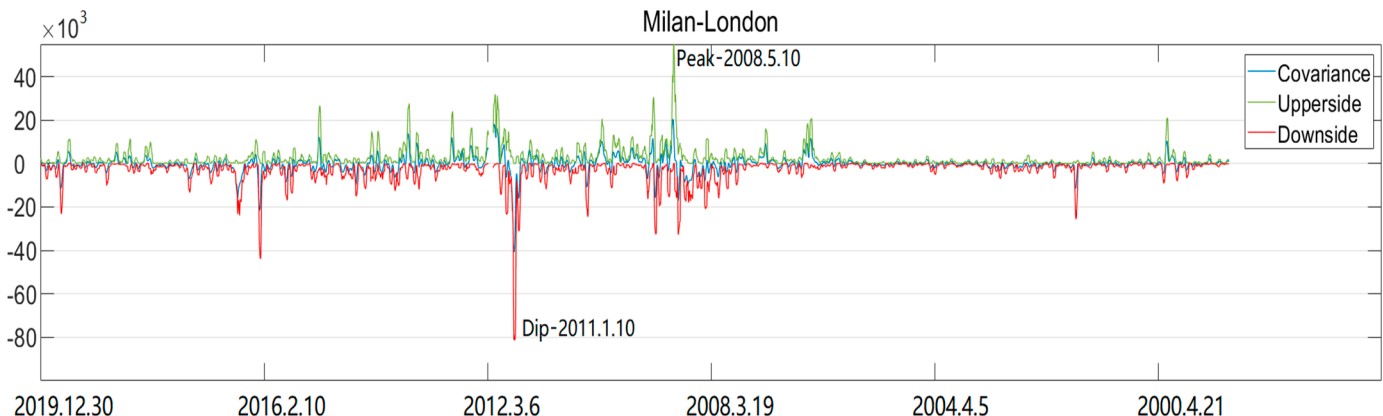

**Figure 2.** Bi-weekly/semi-covariances between gold price and Milan–London Index UK from the year 2019 back to 1999.

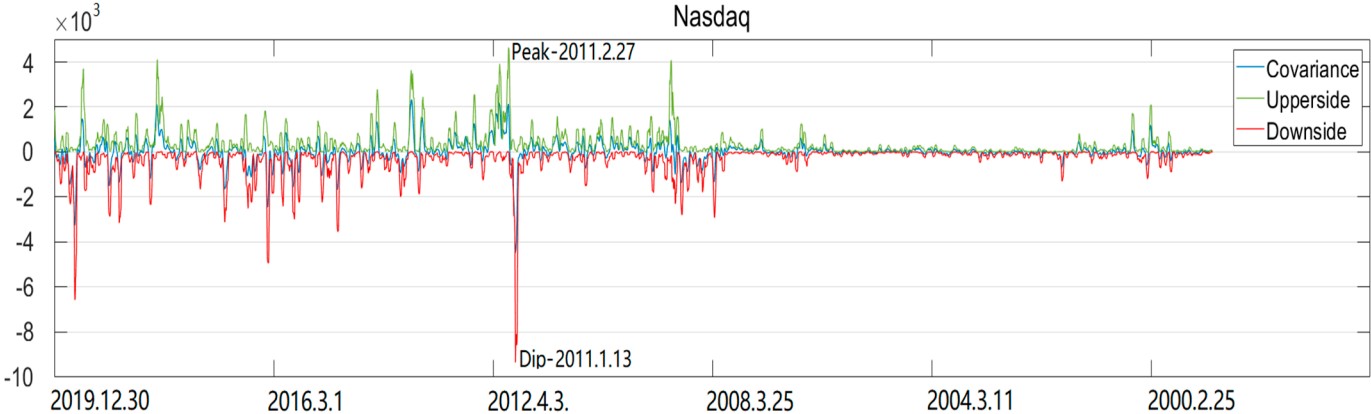

**Figure 3.** Bi-weekly/semi-covariances between gold price and Nasdaq Index USA from the year 2019 back to 1999.

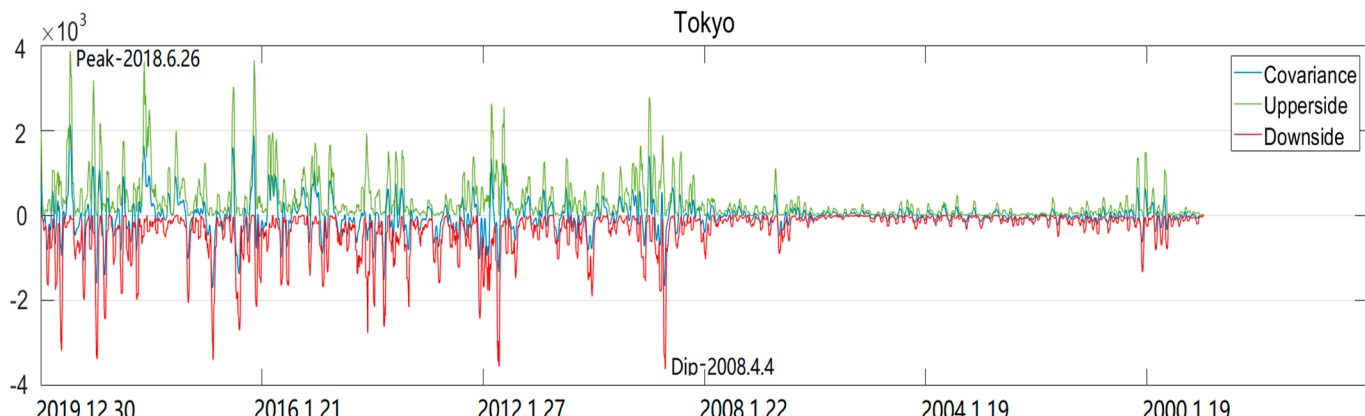

**Figure 4.** Bi-weekly/semi-covariances between gold price and stock for Tokyo Japan from the year 2019 back to 1999.

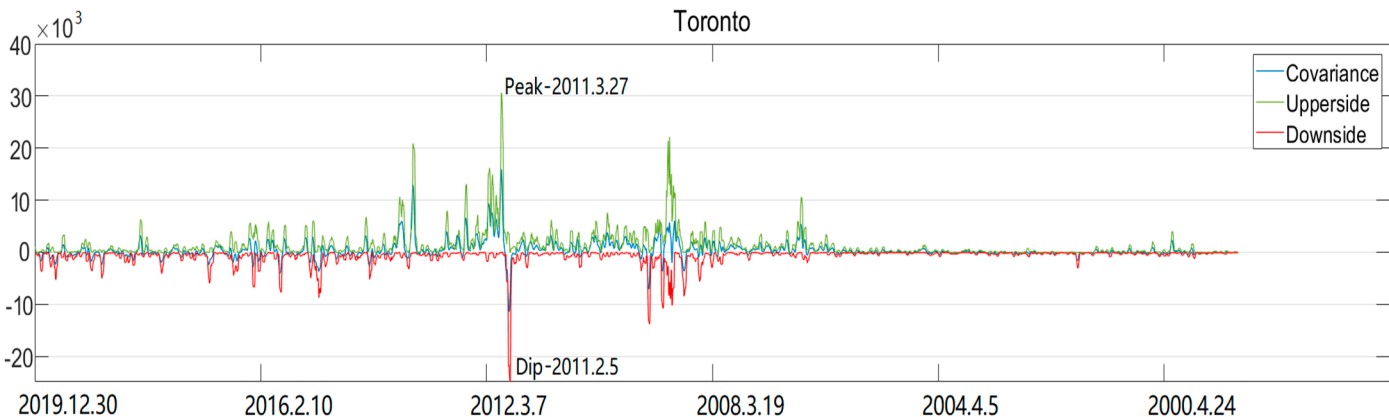

**Figure 5.** Bi-weekly/semi-covariances between gold price and Toronto Index Canada from the year 2019 back to 1999.

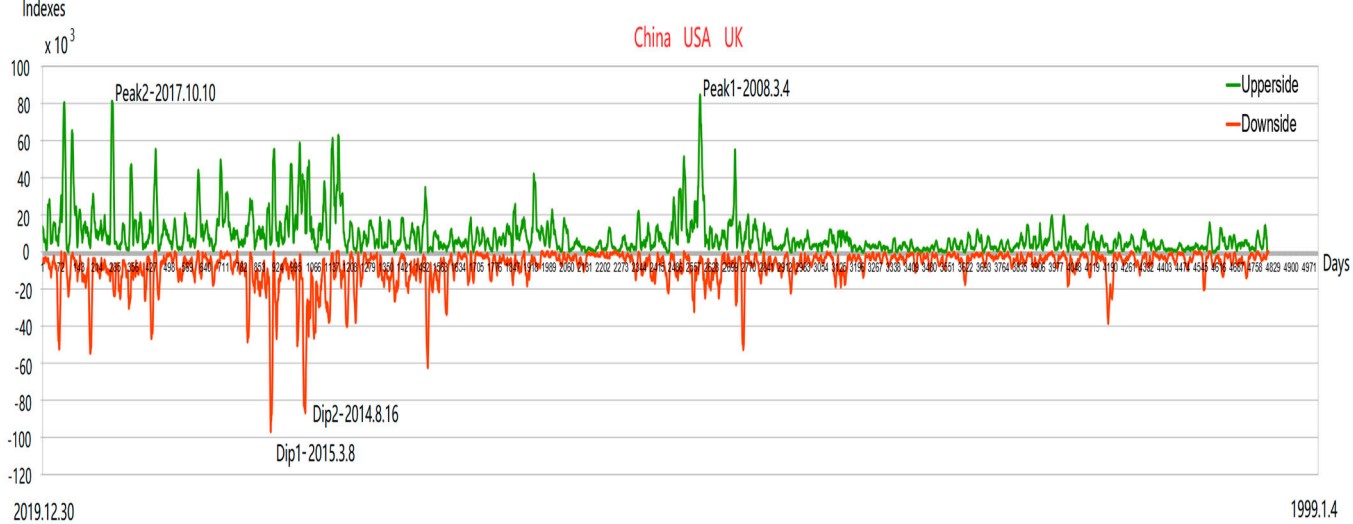

**Figure 6.** Semi-covariances between gold price and China, USA, and UK from the year 2019 back to 1999.

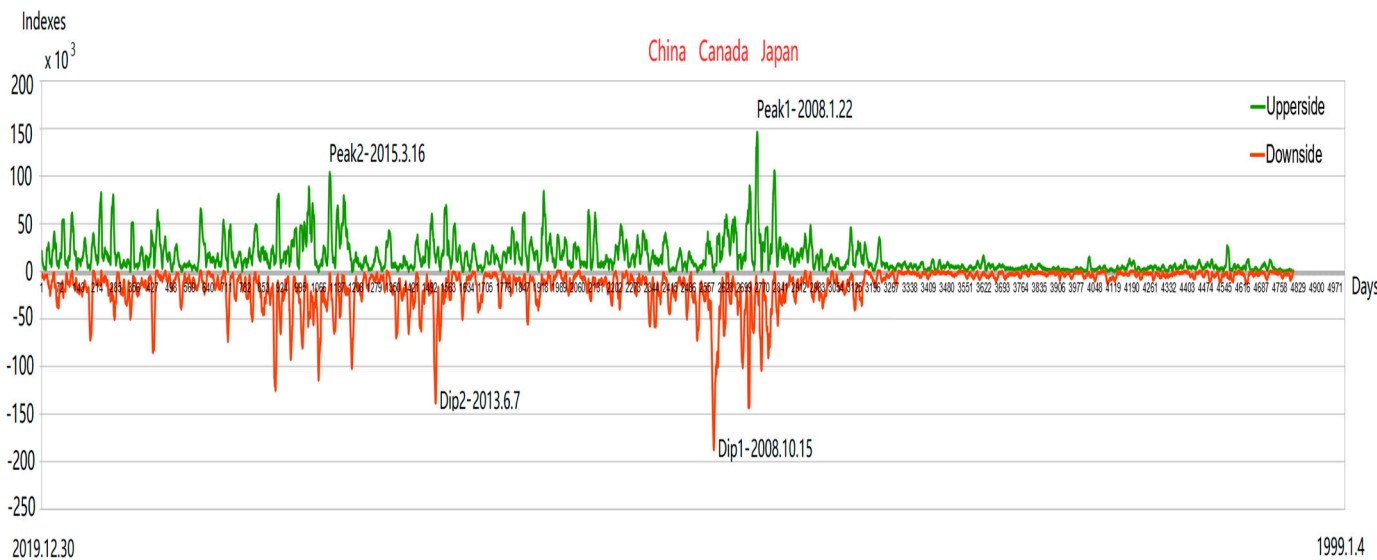

**Figure 7.** Semi-covariances between gold price and China, Canada, and Japan from the year 2019 back to 1999.

**Table 1.** 2019–2009 and 2009–1999 years' gold price in USD to stock index correlations.

|  | Covariance 2019–2009 | Covariance 2009–1999 | Upperside 2019–2009 | Upperside 2009–1999 | Downside 2019–2009 | Downside 2009–1999 |
|---|---|---|---|---|---|---|
| China | −1413 | 3346 | 10,153 | 17,354 | −16,601 | −7994 |
| UK | −22,931 | −5732 | 65,613 | 68,596 | −119,844 | −93,950 |
| USA | 248 | 1036 | 16,921 | 6265 | −13,190 | −3980 |
| Japan | 542 | 772 | 16,866 | 5172 | −13,402 | −3808 |
| Canada | 11,846 | 8807 | 48,741 | 28,515 | −21,187 | −8957 |
| Average | −2342 | 4115 | 31,659 | 25,180 | −36,845 | −23,738 |
| Gold/China | /UK/USA | 4 variables: | 11,560 | 5892 | −10,341 | −5608 |
| Gold/China | /Can/Jap | 4 variables: | 19,284 | 9736 | −19,410 | −11,798 |

Figure 2 shows the correlation between the London stock market and the price of gold. The chart shows that the relationship between the London Stock Index and the price of gold was weak in the early periods 1999–2019. By comparing the above and below covariance, we could see that the London stock market has a relatively better negative correlation with gold prices. In addition, it seems that time factors have a greater impact on precious metals and the long-term UK stock market. In other words, gold cannot always be used as a hedge and a safety net for the stock market, but buying gold and holding for a longer period of time may be a right choice; especially when the UK stock market has experienced extreme negative shocks from specific stock events. The negative peak day on this curve is 10 January 2011. Before this day, the London stock market had risen for four consecutive days. According to behavioral psychology, perhaps people consider that their stock will fall with high probability on this day. Therefore, they withdraw funds from the stock market, which makes the gold market form a safe position. The conclusion is similar to Nguyen and Bhatti's study in 2016.

Figure 3 illustrates the relationship between the Nasdaq stock market and the price of gold. In this figure, the Nasdaq stock market index and gold prices had more upside covariance than downside covariance, but the value of the below mean is higher than the value of the above mean. Although the numerical results displayed are less relevant than the UK, the covariance between the index of US equities and gold prices fluctuates more regularly. In which case, gold might be a safety net for US financial transactions. The positive peak day on this curve is 27 February 2011. The negative peak day on this curve is 13 January 2011. The relationship between the Nasdaq stock market and the gold market has changed from a very positive correlation to a very negative correlation in a

month. The reason is that the stock market has experienced substantial volatility during this period, which has caused market funds to be updated. The appearance of these two peaks may also herald the bull market in the next ten years. The result is the same as Baur and McDermott's study in 2010.

Figure 4 displays the correlation between the Tokyo stock market and the price of gold. By comparison, we could find that the fluctuation pattern of covariance in Tokyo is similar to that of the United States in general. The difference between the two markets is that, the absolute value of the covariance of the U.S. stock market will be greater at some particular point, which further suggests that buying gold as a hedge in the Japan financial market is a good option when the stock market faces frequently negative shocks. The positive peak day on this curve is 26 June 2018. The negative peak day on this curve is 4 April 2008. One of the factors affecting the Japanese stock market may be Japan's geographical environment, such as earthquakes.

Figure 5 shows the covariance between the market value of Toronto's stock and the price of gold. The correlation between Toronto's stock market index and precious metals prices is similar to China's because the covariance values are comparable in the long run. In addition, the chart shows that the Canadian stock market and gold prices are positively correlated in 2019–1999. This represents that gold may not be an effective hedge in Canada's financial market. The positive peak day on this curve is 27 March 2011. The negative peak day on this curve is 5 February 2011. Compared with those days in the USA chart, we can see the Canadian market is around a month behind USA. This conclusion confirms the findings of Baur and McDermott's study in 2010.

Figure 6 illustrates the relationship between the comprehensive stock market and gold prices in China, the United States, and the United Kingdom. The four variable upside covariance is similar to the fluctuation pattern and values of the four variable downside covariance. Therefore, we might suppose that the market index of the three dominant countries is more relatively relevant to the price of gold gradually. In other words, the gold market may not be suitable as a hedge and safe haven for the joint stock markets.

Figure 7 illustrates the relationship between the comprehensive stock market and gold prices in China, Canada, and Japan. The four variable upside covariance is similar to the fluctuation pattern and values of the four variable downside covariance. Therefore, we might suppose that the market Index of the three key countries is less relatively relevant to the price of gold gradually. In other words, the gold market may be suitable as a hedge and safe haven for the joint stock markets.

From the above 7 charts, we can see that Pearson covariance only reflects the two variables variation after the cancellation of up and down, semi-covariance reflects the direction of the variations before the cancellation of two or even four variables as well.

From the above table, we can see that complex variances reveal more dependency and trend for each country's (or the group of countries) economy developing pace.

Table 1 displays the numerical covariances of the five countries and the leading countries selected for drawing Figure 6. In addition, this table separates the time series into two periods, 1999–2009 and 2009–2019. 2009 is a financial crisis year. This way we see the difference before and after the crisis. As seen from the table, the market index of stocks in China, the United States, Japan, and Canada were positively related to the price of gold in the first ten years. Japan had the weakest correlation, while Canada had the strongest. From 2009 to 2019, China's covariance had changed from positive to negative. The negative correlation between the market value of UK stocks and the price of gold has expanded further, mainly because the UK stock market may experience extreme negative shocks over a short period of time. Gold may be a good option during this period, but it does not mean that this has always been the case. During these two periods, Japan's overall covariance was stable and the correlation was not large enough, perhaps a powerful method of hedging equity risk is gold in Japan's financial market. In the first decade, the global stock market is positively related to gold prices and in the second decade it is negative. However, that does not mean gold is an effective hedge for most countries, and

we should apply specific analysis for different situations. Table 2 shows the rank of the safe cycle. China is the shortest, Japan is the longest. Four variable cycle is longer than two variable cycle, because the more markets involved, the less likely they are to be in synchronization.

**Table 2.** 2019–1999 years' gold price in USD to stock index correlation cycle.

|  | Positive Length | Negative Length |
|---|---|---|
| China | 60 | 41 |
| Canada | 98 | 46 |
| USA | 73 | 49 |
| UK | 73 | 73 |
| Japan | 58 | 77 |
| Average | 72 | 57 |
| CH/UK/US | 107 | 117 |
| CH/CA/JP | 137 | 130 |
| Average | 122 | 124 |

## 4. Conclusions

This study constructed a complex covariance for fractional analysis of gold stock market fluctuations using fractional moment basis with a simple algorithm coded in Excel, other similar tools can also be used. This novel complex model reveals extra performance index over the traditional two variable model. The study also provided graphical comparisons of the traditional calculation of the Pearson dimension with that of the fractional dimension for two variable cases (for four variable cases there is no such thing called Pearson, no comparison is given here). The results of the complex calculation show the difference between high frequency trading and low frequency trading that traditional covariance definition and calculation may overlook. It reveals the unique multi-factorized correlation among the price of gold and the stock market for countries.

It is envisioned that the complex model is a good alternative in the stock market prediction or risk analysis due to its asymmetrical fluctuation to help develop high dimensional Fractal theory and easier practical applications. The simplified multi-factorized Excel is an effective calculation tool, accurate enough and backward compatible with traditional two variable Pearson model and calculations. The example code is available from Excel file (accessed on 31 March 2021) on the github server (https://github.com/steedhuang/Gold-price-to-Nasdaq-Shanghai, 31 March 2021) or mirror server (https://gitee.com/steedhuang/gold-price-to-nasdaq-shanghai, 31 March 2021). Our future work will examine the difference among this complex covariance, the traditional realized matrix covariance, modern AI (LSTM model for stock value prediction or BERT model for social media data gathering) based partial covariance, and the pure analytic Lotka–Volterra dynamical model. The multi-factorized semi-covariance theory is still in its infancy, and the principles of more physical world can be explored by this research method in the future.

**Author Contributions:** Conceptualization-drafting, J.S.H. and Y.S.; formal analysis—review and editing, L.Y. and M.H. All authors have read and agreed to the published version of the manuscript.

**Funding:** This research was partially funded by National Natural Scientific Foundation of China Grant Number 11871435. The APC was funded by Jiangsu University.

**Institutional Review Board Statement:** Not applicable.

**Informed Consent Statement:** Not applicable.

**Data Availability Statement:** Public day closing data on stock market.

**Acknowledgments:** We would like to express our sincere gratitude to Xu Ding Hua, for his helpful suggestions in completing this paper. Thanks also go to Tong Xu for drawing the figures. In addition, the research is partially supported by the National Natural Scientific Foundation of China (Grant No. 11871435).

**Conflicts of Interest:** The authors declare no conflict of interest.

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
