# Peer review of "Multi-Factorized Semi-Covariance of Stock Markets and Gold Price"

_jrfm, doi:10.3390/jrfm14040172_

Round 1

Reviewer 1 Report

This is an interesting study of the semi-covariance between the gold market and various stock markets around the world. It finds that the relationship between the markets is complex but at times gold is a suitable hedge. I have the following comments:

  • In the introduction the authors could state the specific contributions of the paper to the literature.
  • The paper could add additional literature on the study of covariances between markets.
  • In section 2, the equations need to be numbered and explained in more detail and the variables defined.
  • In the figures, they need to be bigger to make them clearer and also the dates need to be added on the horizontal axis rather than numbers.
  • In Table 1, why was the data split in 2009, was this the financial crisis?
  • Overall the results need to be compared to other similar studies' results on these markets.

Minor Points:

  • A brief overview needs to be added at the end of the introduction.
  • The references need to be consistent, i.e. some have dates after authors, some are at the end of the reference.

Author Response

  • In the introduction the authors could state the specific contributions of the paper to the literature.

Response 1: We have added the purpose of writing the article in the part of introduction and emphasizing the contribution of this paper to financial world.

  • The paper could add additional literature on the study of covariances between markets.

Response 2: We have added several literatures about the researches around the covariances between different countries’ markets before the statement.

  • In section 2, the equations need to be numbered and explained in more detail and the variables defined.

Response 3: We numbered the equations and defined the variables in more detail, which make it easier for readers to understand.

  • In the figures, they need to be bigger to make them clearer and also the dates need to be added on the horizontal axis rather than numbers.

Response 4: The figure has been enlarged to make it clearer, and its horizontal axis has also been added dates.

  • In Table 1, why was the data split in 2009, was this the financial crisis?

Response 5: Yes, the reason why the data divided on 2009 was the financial crisis. 1999 to 2009 is 10 years, and 2009 to 2019 is another 10 years. It is a fair comparison before and after for the same length.

  • Overall the results need to be compared to other similar studies' results on these markets.

Response 6: We compare the results of this article with the traditional Pearson model. In addition, we also compare the results of the analysis with previous studies in the specific analysis of each country.

  • A brief overview needs to be added at the end of the introduction.

Response 7: We added a brief overview at the end of the introduction with contributions of this paper.

  • The references need to be consistent, i.e. some have dates after authors, some are at the end of the reference.

Response 8: The format of references has been revised in accordance with the standard of the journal’s template.

Reviewer 2 Report

The paper "Multi-Factorized Semi-covariance of Stock Markets and Gold Price" by Yun Shi, Lin Yang, Mei Huang and Jun Steed Huang adresses an interesting topic. It follows the usual pattern with which I strongly agree. It is very synthetic and complete as well. All the section have enough content of high quality. The quality of the English writing is very good. Nevertheless, I recommend an attentive new reading. The Nasdak word was changed once to Dasdak, so it needs to be rectified. I propose to publish the article after a new and carefull reading.

Author Response

  • Nevertheless, I recommend an attentive new reading. The Nasdak word was changed once to Dasdak, so it needs to be rectified. I propose to publish the article after a new and carefull reading.

Response: The word Dasdak has been changed to Nasdak, the rest parts are all double checked again.

Reviewer 3 Report

General comment

The paper in general is interesting. The authors propose  a complex models used to explain the stochastic phenomenon with upward and downward fluctuation such as the stock market. They propose a simplified multi-factorized fractional dimension derivation with the exact Excel tool algorithm involving the fractional center moment extension to covariance, which is a complex parameter average that is a multi-factorized extension to Pearson covariance.

Specific comment

The authors introduce well the issue addressed in their paper in the Introduction section, but do not clearly set in that section its objective, nor how it will be developed to follow.

In section 2.Sub, Super or Pearson Covariances, the authors clearly introduce the methodological approach.

In section 3. Excel Calculations of Semi-covariances among the Price of Gold and Stock, the authors highlight the results for different markets and geographical areas. The description of the results is quite clear, but I invite them to provide more information about the peak day for the different cases analyzed. In fact, in this case it is not possible to relate their analysis with this information which is important enough to have a complete knowledge framework.

In section 4. Conclusion, the authors propose a summary of the results obtained and some considerations on the algorithm used. In this section, however, the authors do not identify the possible developments of this research.

In addition, the authors state that “The example code is available from Excel file on the Research gate server under the last author profile”, this is indeed true, but it is advisable to insert this code in the Appendix to this article and in English, since the one on Research gate is in Chinese, in order to complete the knowledge of the model in the context in which they are proposing it.

Author Response

  • The authors introduce well the issue addressed in their paper in the Introduction section, but do not clearly set in that section its objective, nor how it will be developed to follow. In section 2.Sub, Super or Pearson Covariances, the authors clearly introduce the methodological approach.

Response 1: We have added the objective of this paper in the part of introduction, and described how this article was developed to follow.

  • In section 3. Excel Calculations of Semi-covariances among the Price of Gold and Stock, the authors highlight the results for different markets and geographical areas. The description of the results is quite clear, but I invite them to provide more information about the peak day for the different cases analyzed. In fact, in this case it is not possible to relate their analysis with this information which is important enough to have a complete knowledge framework.

Response 2: We have provided additional information about peak days such as the reason for its occurrence or major events that may affect the stock market.

  • In section 4. Conclusion, the authors propose a summary of the results obtained and some considerations on the algorithm used. In this section, however, the authors do not identify the possible developments of this research.

Response 3: In the conclusion section, we provided the research-ability of Semi-Covariance theory of this article in the future and the applicability of this theory in the real world.

  • In addition, the authors state that “The example code is available from Excel file on the Research gate server under the last author profile”, this is indeed true, but it is advisable to insert this code in the Appendix to this article and in English, since the one on Research gate is in Chinese, in order to complete the knowledge of the model in the context in which they are proposing it.

Response 4: The Excel files of this paper is too large, which may not be insert in the Appendix. The document has been updated from Chinese to English and moved to more popular github server, and the Uniform Resource Locator has been inserted at the end of the paper.

Round 2

Reviewer 3 Report

Authors have improved the level of their paper by integrating the sections in which critical issues were highlighted, now this product can be published in JFRM Journal